# NextVir: Enabling classification of tumor-causing viruses with genomic foundation models

**John Robertson** *, **Shorya Consul** , **Haris Vikalo**

Chandra Family Department of Electrical and Computer Engineering, The University of Texas at Austin, Austin, Texas, United States of America

* john.robertson@utexas.edu

## Abstract

**Motivation:** Oncoviruses, pathogens known to cause or increase the risk of cancer, include both common viruses such as human papillomaviruses and rarer pathogens such as human T-lymphotropic viruses. Computational methods for detecting viral DNA from data acquired by modern DNA sequencing technologies have enabled studies of the association between oncoviruses and cancers. Those studies are rendered particularly challenging when multiple species of oncovirus are present in a tumor sample. In such scenarios, merely detecting the presence of a sequencing read of viral origin is insufficiently informative—instead, a more precise characterization of the viral content in the sample is required.

**Results:** We address this need with NextVir, to our knowledge the first multi-class viral classification framework that adapts genomic foundation models to detecting and classifying sequencing reads of oncoviral origin. Specifically, NextVir explores several foundation models—DNABERT-S, Nucelotide Transformer, and HyenaDNA—and efficiently fine-tunes them to enable accurate identification of the sequencing reads' origin. The results demonstrate superior performance of the proposed framework over existing deep learning methods and suggest downstream potential for foundational models in genomics.

## Author summary

Cancer-causing viruses, known as oncoviruses, are responsible for approximately 15% of human cancers worldwide. Detecting and identifying these viruses in tumor samples is critical for understanding how cancers form and for developing more effective treatments. In this study, we introduce NextVir, a new artificial intelligence (AI) tool that not only detects the presence of DNA viruses in a sample but also identifies which oncoviral family it belongs to. This distinction is important because tumors may involve

**Data availability statement:** Code is available at https://github.com/johntrob14/NextVir, where there are also links to the dataset and models on Zenodo.

**Funding:** This work was supported in part by the National Science Foundation (nsf.gov) grant #2109983, which was awarded to HV and used as salary for JR. The funder played no role in study design, data collection and analysis, decision to publish, or preparation of the manuscript.

**Competing interests:** The authors have declared that no competing interests exist.

multiple viruses simultaneously, and the interactions between them are poorly understood. NextVir adapts large AI models originally designed for general genomic analysis, to the specific task of viral detection. Our experiments show that NextVir is accurate and robust, even in challenging settings that emerge due to various sequencing artifacts. NextVir outperforms many standard methods on viral detection tasks, demonstrating the effectiveness of adapting general genomic models to specific use cases. By improving our ability to detect and classify viruses that contribute to cancer, NextVir offers a powerful new approach to gain a deeper understanding of tumor development mechanisms.

## Introduction

Several viruses are well-known to be associated with human cancers, either increasing the risk or downright causing the disease [1,2]. Examples of such so-called oncoviruses include Human papillomaviruses (HPVs), linked with cervical cancer; Epstein-Barr virus (EBV), known to promote certain lymphomas; and Hepatitis B virus (HBV) and hepatitis C virus (HCV), tied to liver cancer. This has motivated research efforts to improve our understanding of the molecular mechanisms of viral carcinogenesis [3]. For instance, it is known that oncoviruses encode viral oncoproteins, which may impact the regulatory cellular processes in the host, leading to tumor formation. In addition to this, viral genetic material integrate into the host genome, which may in turn confer a proliferative advantage on infected cells and result in tumor formation.

Studies of viral carcinogenesis mechanisms are predicated upon our ability to map the oncoviral landscape, i.e., identify viral genomic content in tumor cells. This task has proven to be challenging due to the incompleteness of viral genome databases, coupled with the rapid mutations exhibited by oncoviral families. Moreover, even low amounts of viral content may prove to be causative for cancer [4], necessitating reliable classification capabilities. Most existing approaches for viral detection from RNA, cDNA and/or peptides seek to frame this problem as a binary classification, wherein the goal is to simply identify whether a given sequence is of viral origin. Examples of such methods include ViFi [5], viRNAtrap [6] and DeepViFi [7]. ViFi constructs an ensemble of Hidden Markov Models (HMMs) built from viral reference genomes to identify viral reads that may have evolved from those genomes. In contrast, viRNAtrap employs a convolutional neural network (CNN) to identify relatively short viral RNA sequencing reads, while DeepViFi utilizes a combination of a transformer and random forest for viral DNA read identification. Deep learning methods for similar problems in metagenomics include the CNN-based DeepVirFinder [8] and LSTM-based Virtifier [9].

While all of the aforementioned techniques focus on binary classification of pathogens, recent studies have revealed that tumors can exhibit multiple viral infections [10–12]. In such settings, the existing methods are limited to detecting the presence of viral genomic material in a sample but are unable to ascertain the virus that the sequence originated from. A potential solution based on training a set of binary classifiers, one for each class of oncoviruses, requires high computational and memory costs, and is time-consuming. This motivates our pursuit of multi-class viral classifiers built atop of genomic foundation models: since they are trained to provide "good" general-purpose latent representation of genomic sequences, such models appear inherently felicitous for the task of distinguishing between components of a viral mixture.

## Foundational models in genomics

DNA sequences that may be used in viral classification tasks (and, more broadly, other tasks in genomics) vary significantly in size—short and long sequencing reads, contigs and scaffolds, complete genome assemblies, etc. Existing deep learning models developed for such tasks utilize various techniques to encode the genetic information and relationships inherent to these sequences. A natural starting point is to use each nucleotide base as a token, yielding a vocabulary comprising only four tokens corresponding to adenine, cytosine, guanine and thymine; HyenaDNA [13] opts for such a tokenization to preserve single-nucleotide resolution. Other approaches, such as DNABERT [14] and Nucleotide Transformer [15], tokenize the sequence into k-mers, i.e., subsequences of length k. On one hand, such a tokenization is advantageous as it confers each token with some contextual information. On the other hand, k-mer driven tokenization may suffer from poor sample efficiency due to information leakage when using overlapping k-mers, and the sensitivity to insertions and deletions when using non-overlapping k-mers. More recent foundational models, including DNABERT-2 [16] and DNABERT-S [17], elect to tokenize genomic sequences using SentencePiece [18] with Byte Pair Encoding [19]; this tokenizer dispenses with any predefined notion of a 'word' and instead seeks to construct a fixed-size vocabulary based on co-occurrence frequencies of the bases. Next, we briefly overview the considered foundation models.

**DNABERT-S.** DNABERT-S gains the ability to distinguish between genomic material originating from different species by refining its predecessor, DNABERT-2, through a two-step training strategy dubbed $C^2LR$ or Curriculum-Contrastive Learning. The first step relies on Weighted SimCLR [20] to learn sequence embeddings. The second step utilizes the Manifold Instance Mixup (MI-Mix) method to create progressively more challenging positive and negative pairs of embeddings that the model should distinguish between. The dataset for the aforementioned training comprises pairs of non-overlapping DNA sequences, each 10k base pairs (bp) in length, originating from the genomes under consideration. One of the sequences in each pair is arbitrarily fixed as the anchor to reduce computational complexity.

**Nucleotide transformer.** The Nucleotide Transformer consists of a series of foundational models of varying sizes, ranging from 500 million to 2.5 billion parameters. These models employ a BERT-style encoder in order to learn generalizable embeddings from a diverse collection of genomes using masked language modelling (MLM). In particular, the Nucleotide Transformer tokenizes each input sequence as a series of non-overlapping k-mers (a mix of 6-mers and 1-mers). For our experiments,, we focus on the 500 million parameter model, as it is the closest in parameter count to the other foundational models we considered.

**HyenaDNA.** In contrast to the aforementioned models, HyenaDNA is a decoder-only architecture trained for next-token prediction on the human reference genome. It leverages the Hyena operator [21] to greatly expand the context length of the traditional transformer block from 4096 tokens to 1 million tokens. In addition to this, it forgoes the use of a tokenizer and encodes each sequence using the minimal DNA vocabulary consisting of 4 letters.

## Contributions

We present a novel deep learning framework, NextVir, for detecting and classifying DNA sequencing reads of oncoviral origin. NextVir builds on three foundation models by fine-tuning their embeddings using the low-rank weight matrix adaptation (LoRA) technique [22] and adding an adapter network on top to accurately classify input sequencing reads based on their origin. The remainder of the paper is organized as follows. In Methods, we outline the NextVir framework and provide details of the network architecture. In Results, we

demonstrate the ability of NextVir models to attribute reads to oncoviral families on semi-experimental data. We then proceed to compare the performance of NextVir to state-of-the-art methods at identifying sequencing reads of viral origin. Finally, Discussion recounts the paper and outlines future work directions.

## Methods

### Problem formulation

High-throughput sequencing platforms are capable of generating massive amounts of sequencing reads characterizing genomic material in a biological sample [23]. Each such read can be thought of as being obtained by sampling, with replacement, the genomes present in the sample. When sequencing genetic material collected from tumor cells, an oncoviral infection is manifested via the presence of sequencing reads originating from the infecting virus. Formally, the classification problem that we pursue can be stated as follows: *For each sequencing read $r_i$, can we determine whether it is of viral origin and, if so, which oncoviral family $y_i$ it belongs to?* As argued in Introduction, prior works are restricted to binary classification settings where $y_i$ is constrained to be binary (with 0 indicating non-viral and 1 indicating viral origin). For our problem, on the other hand, $y_i \in \{0, 1, 2, ..., n\}$; here 0 still indicates non-viral, i.e., human origin, while the other values of $y_i$ indicate different oncoviral families. We will focus on optimizing NextVir models for the multi-class case, however we will also train and test binary models for comparison with existing viral classification methods.

### NextVir framework

The architecture of NextVir is illustrated in Fig 1. It comprises two components - a foundational base model and an adapter network. The foundational base models employ different architectures and tokenization schemes, but have each been shown to perform well on a variety of downstream genomic tasks. We train and test variants of NextVir built atop each of the three discussed foundational models: DNABERT-S (leading to NextVir-D), Nucleotide Transformer (NextVir-N), and HyenaDNA (NextVir-H). The sequence embeddings from each base model are mean-pooled prior to the adapter network, which computes the output logits.

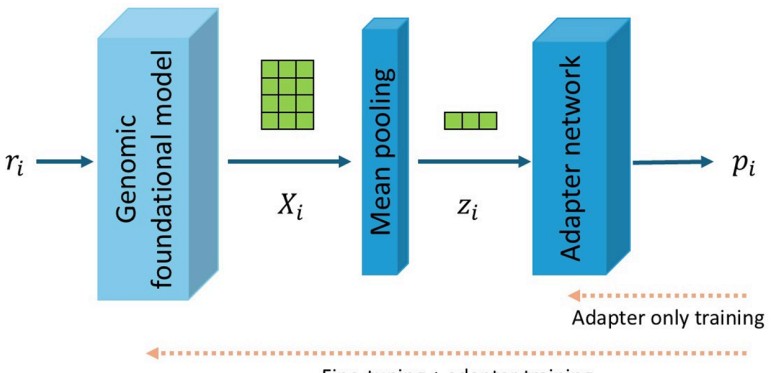

**Fig 1. An illustration of the NextVir framework.** The dashed arrows indicate the layers affected by the various training regimens.

**Input preprocessing.** NextVir models are trained on next-generation sequencing (NGS) reads of length 150 base pairs (bp), typical of Illumina's sequencing platforms. This stands in contrast to the three foundational models: DNABERT-S was pre-trained on 700bp long sequences and fine tuned using sequences 10,000bp in length, HyenaDNA saw full reference genomes of up to 1 million bp, while NucleotideTransformer was trained on 1000bp long sequences. Note that DNABERT-2 (the predecessor of DNABERT-S) was not shown any viral sequences while the fine-tuning data used to create DNABERT-S contained only small amount of viral data (2.34% of the dataset). Similarly, HyenaDNA and Nucleotide Transformer saw no viral DNA during pre-training; instead, they were trained on a human reference genome and a collection of species excluding plants and viruses, respectively. We opt to train NextVir models on short NGS reads due to their prevalence in sequencing studies of tumor samples.

Each read is tokenized according to the tokenization scheme of the corresponding base model. In the case of NextVir-D, this results in inputs of different lengths. To enable efficient batch processing for such sequences, all sequences are padded to a uniform length by appending special padding tokens to sequences shorter than the maximum length in the batch. An accompanying attention mask is applied to each input to ensure that padding tokens are ignored during attention computations. This mechanism ensures that padding does not influence the model's predictions. Padding enables the simultaneous processing of variable-length reads in a batch, which is essential for the practical use of transformer models. Without padding, training would be restricted to one read at a time, resulting in prohibitively high computational costs.

**Pooling.** Recall that after tokenization, each read is represented as a sequence of $k$ tokens. The corresponding transformer model encodes this sequence into a $k \times L$ embedding matrix, where $L$ denotes the dimensionality of the latent space: 768, 1024, and 256 for NextVir-D, NextVir-N, and NextVir-H, respectively. To obtain a fixed-size representation for downstream classification, we apply mean pooling across the token dimension, resulting in an $L$-dimensional vector for each input sequence. While more sophisticated pooling strategies (e.g., attention-based or learned linear pooling) could potentially further enhance the model performance, mean pooling offers a parameter-free alternative with lower computational complexity and training cost. Despite its simplicity, this approach allows NextVir to maintain strong classification accuracy.

**Adapter.** Each NextVir model employs a lightweight two-layer feedforward adapter to compute class logits from the pooled sequence embeddings. Specifically, the adapter consists of two fully connected layers with a ReLU activation between them. Given a pooled embedding vector $z_i \in \mathbb{R}^L$, the adapter computes the output according to

$$FFN(z_i) = max\left(0, z_i W_1^{ff} + b_1^{ff}\right) W_2^{ff} + b_2^{ff}, \tag{1}$$

where $W_1^{ff}$ and $W_2^{ff}$ are the weight matrices of the feedforward layers, and $b_1^{ff}$ and $b_2^{ff}$ are the corresponding biases.

The latent dimension of the adapter is 64, i.e., the adapter projects each $L$-dimensional input into a 64-dimensional latent space before producing the final output logits. In the multiclass classification setting, the output dimension $C$ is equal to the number of target viral families plus one, accounting for the non-viral class. The final class probabilities are obtained by applying the softmax function to the output logits.

## Network optimization

DNABERT-S, Nucleotide Transformer, and HyenaDNA comprise in excess of 100 million, 450 million, and 6.5 million parameters, respectively. Fine-tuning such large-scale models presents a significant computational challenge. To mitigate this, we train the self-attention matrices (query, key and value matrices) and the dense layers of each model by relying on the low-rank weight matrix adaptation (LoRA) technique [22]. LoRA posits that the weight update has low "intrinsic rank", so it learns low-rank matrix decompositions of the weight updates. These weight updates can then be added to the weights of the foundation model to obtain fine-tuned model weights. Since the number of learnable parameters scales linearly with rank, for computational efficiency, LoRA updates in NextVir are generated using an intrinsic rank of 4. While larger ranks may perform marginally better, as shown in S1 Appendix, they introduce longer training times and require large computational costs to ensure convergence to a fully fine-tuned model.

**Loss function.**   Training of NextVir aims to minimize weighted cross-entropy loss formally defined as

$$L = -\sum_{i=1}^{M} \frac{M}{M_{y_i}} y_i \log(p_i),\qquad(2)$$

where $M$ denotes the total number of reads and $M_j$ denotes the number of reads in the $j^{th}$ class. This loss function upweights accurate classification of reads belonging to classes where $M_j$ is small, i.e., the classes that are present in the training set at low abundances. In effect, this encourages the model to classify all the classes equally well.

**Optimization and hyperparameter tuning.**   All NextVir variants are trained using the schedule-free AdamW optimizer with a learning rate of 0.001, $\beta = 0.85$ and $\lambda = 0.005$. Hyperparameter selection is performed via two independent grid searches on the validation set. The first search focused on identifying optimal learning rates in the range [0.0005, 0.005] for both NextVir-D fine-tuning and adapter training. The second search targeted optimal values for $\beta_1$ and $\lambda$ within the ranges [0.85, 0.95] and [0.001, 0.01], respectively. Further details on the grid search procedures can be found in S1 Appendix. All models are trained for 15 epochs, with 40% of the first epoch used as warmup steps. Model selection is based on validation loss.

**Hardware and software.**   All experiments are conducted on a shared Linux server equipped with an AMD EPYC 7642 48-core CPU and 512 GB of DDR4 memory. Training the NextVir multi-class model takes 12 hours using 4 GPUs for 15 epochs. Experiments are spread across 8 AMD Vega 20 32GB GPUs with RoCM. The code is implemented in Python, using PyTorch-ROCm for distributed processing. The codebase will be made publicly available. DNABERT-S and AdamWScheduleFree are used in accordance with their Apache 2.0 licenses. The DNABERT-S weights used are available at https://huggingface.co/zhihan1996/DNABERT-S/commit/1cdf84d992ace6f3e75c7356774b4da088c8dc7c and the optimizer used is available at https://github.com/facebookresearch/schedule_free or through pip as 'schedulefree==1.2.5'. NucleotideTransformer is used in accordance with its Creative Commons License; the weights are available at https://huggingface.co/InstaDeepAI/nucleotide-transformer-v2-500m-multi-species/commit/\protect\penalty-\@M{}f1fd7a1df5b19d31b88f11db1ce87caeb1ea4d2a. HyenaDNA is used in accordance with its BSD 3-Clause license; the weights used are available at https://huggingface.co/LongSafari/hyenadna-large-1m-seqlen/commit/bf15705b2037667c89fbc6aa4126ead6da403bf3. Numerous experiments are run concurrently, some of which have been omitted from the paper for brevity. Across all experiments, an estimated 15 GPU-days of compute are consumed.

## Results

### Experimental setting

We construct our dataset using 18,680 oncoviral genomes from the integrative database of cancer-associated viruses (iCAV) [24]. These genomes span 7 viral families known to include carcinogenic subtypes: Hepatitis B (HBV), Human-Papillomavirus (HPV), Hepatitis C (HCV), Epstein-Barr (EBV), Human T-Cell Lymphotrophic Virus (HTLV), Human Herpesvirus 8 (HHV-8), and Merkel Cell Polyomavirus (MCV). Human DNA sequences are obtained from the primary assemblies of the GRCh38.p14 reference genome [25]. ART [26] is used to generate 150bp Illumina MiSeq reads from all the genomes; this data is then randomly partitioned according to an 80:10:10 split into training, validation, and testing sets, respectively. Table 1 summarizes the genome lengths and the number of sequencing reads generated for each viral family and the human reference.

### Classification with pretrained embeddings

We start by testing the capabilities of NextVir with the three models kept as-is, without any fine-tuning of the base model (see Fig 1). In this configuration, only the adapter network (i.e., the two-layer feed-forward classifier) is trained, while the loss gradients are not propagated through the base model. Thus, the mean-pooled embeddings produced by DNABERT-S, HyenaDNA, and the Nucleotide Transformer are solely the result of their extensive pre-training. As discussed in *Input preprocessing*, viral sequences constituted only a small fraction of DNABERT-S's pretraining data, and none of the pretraining data for HyenaDNA or the Nucleotide Transformer included reads from viral genomes. Hence, the burden of adapting to oncoviral sequence classification falls entirely on the lightweight adapter and the models' latent ability to generalize to previously unseen sequence domains.

Table 2 demonstrates that, even without fine-tuning, adapter-only training enables NextVir to learn discriminative embeddings that support effective oncoviral classification. Notably, the per-class accuracy or recall—a fraction of reads in a given class that are correctly classified—exceeds 70% for most viral families. However, performance of these models degrades on classes associated with longer genomes, particularly Epstein-Barr Virus (EBV) and Human Herpesvirus 8 (HHV-8), both of which have average genome lengths exceeding 100,000 base pairs. These longer genomes result in lower sequencing coverage, i.e., fewer reads per base on average, which limits the amount of information available to the model for learning class-specific features. In contrast, this experiment produces high classification accuracies for Hepatitis B Virus (HBV) and Human Papillomavirus (HPV), which have shorter genomes

**Table 1. Semi-experimental dataset: length of the genomes and the number of 150bp reads originating from each genome in the dataset.**

| Species | Average length (bp) | Number of 150bp reads |
|---|---|---|
| Human | $2.94 \times 10^9$ | 464016 |
| HBV | 3195 | 229057 |
| HPV | 7725 | 149184 |
| HCV | 9310 | 34086 |
| EBV | 168013 | 25975 |
| HTLV | 8663 | 5296 |
| HHV-8 | 135310 | 1959 |
| MCV | 5297 | 1709 |

**Table 2. Classification accuracies on test data for different NextVir variants; "pretrained" indicates no fine-tuning of the foundation model. Five seeded runs are averaged for fine-tuned models to calculate an associated 95% confidence interval.**

| Class | Accuracy (%) | | | | | |
|---|---|---|---|---|---|---|
| | NextVir-D pretrained | NextVir-D | NextVir-N pretrained | NextVir-N | NextVir-H pretrained | NextVir-H |
| Overall Top-1 | 77.40 | 94.69 ± 0.34 | 81.51 | 95.02 ± 0.95 | 58.00 | 89.88 ± 0.24 |
| Human | 71.32 | 91.81 ± 0.52 | 75.35 | 92.19 ± 1.59 | 50.90 | 84.57 ± 0.44 |
| HBV | 87.35 | 99.74 ± 0.05 | 93.36 | 99.80 ± 0.08 | 60.22 | 99.53 ± 0.08 |
| HPV | 87.44 | 95.53 ± 0.52 | 87.55 | 96.94 ± 0.35 | 77.87 | 92.91 ± 0.28 |
| HCV | 70.39 | 95.68 ± 0.66 | 78.26 | 96.55 ± 0.67 | 70.84 | 91.54 ± 0.38 |
| EBV | 49.59 | 95.13 ± 0.56 | 56.55 | 91.03 ± 2.24 | 32.34 | 77.84 ± 2.12 |
| HTLV | 81.02 | 98.18 ± 0.53 | 85.20 | 98.63 ± 0.32 | 72.30 | 97.42 ± 0.32 |
| HHV-8 | 38.60 | 62.33 ± 1.41 | 43.72 | 57.30 ± 2.47 | 28.37 | 54.70 ± 2.46 |
| MCV | 80.71 | 99.05 ± 0.59 | 71.60 | 99.88 ± 0.23 | 57.40 | 98.11 ± 0.43 |

and, consequently, higher read coverage. This trend underscores the importance of sequencing depth in achieving accurate read-level classification, particularly when relying on frozen pretrained embeddings.

## Improvements with LoRA fine-tuning

To improve classification performance beyond what is achievable with frozen embeddings, we fine-tune the base models alongside the adapter network, leveraging the LoRA technique for efficient parameter updates (as described in section Network optimization). We select the model with the lowest validation loss for final evaluation on the held-out test set. Table 2 summarizes the resulting test accuracies, both overall and per class, of the trained model. As can be seen there, fine-tuning leads to substantial performance gains across all viral families, with each model achieving overall test accuracy exceeding 90%. In particular, the classification accuracy on EBV reads nearly doubles, reaching levels comparable to more prevalent classes such as HPV and HCV. Adapted models also classify HHV-8 reads much more accurately, with the recall exceeding 60% when using DNABERT-S or Nucleotide Transformer. HyenaDNA struggles the most with this class, but still manages to nearly double its pretrained performance. The reads originating from HHV-8 remain the most difficult to identify. We attribute this difficulty to its extreme rarity in the dataset: HHV-8 reads are over 10 times less abundant than EBV reads (see Table 1), leading to markedly lower sequencing coverage and reduced signal for learning discriminative features.

## Context-supported classification

The results presented thus far suggest that accurate classification of sequencing reads is predicated upon genome-wide high sequencing coverage, i.e., the presence of a sufficient number of reads distributed across the full length of each genome. To further investigate this dependency, we design an experiment in which the training, validation, and test reads originate from non-overlapping genomic regions. This setup simulates realistic conditions in sequencing workflows, such as those involving Polymerase Chain Reaction (PCR), where primer binding may preferentially amplify specific regions of a genome. In extreme cases, this may result in uneven or localized coverage, with some sections of the genome underrepresented or even absent.

We refer to this experimental task as 'context-supported classification'. To implement it, we first align the sequencing reads to the human and viral genomes used to create the dataset

(see section Experimental setting) in order to determine the positions of the reads along their respective genomes of origin. For each class, reads are ordered by position and partitioned such that the first 80% are used for training, the next 10% for validation, and the final 10% for testing. This ensures that each split contains reads from distinct, non-overlapping genome segments. Importantly, validation and test reads do not share context with the training data. A detailed description of this partitioning scheme, along with a visualization, is provided in S1 Appendix.

As expected, the results reported in Table 3 indicate that the overall accuracy for context-supported classification decreases as compared to the setting of Improvements with LoRA fine-tuning (from 94.79% to 86.85% for NextVir-D). Nevertheless, the accuracy in the more challenging context-supported classification setting is significantly higher than the classification with pre-trained embeddings in the previous (easier) setting for all the NextVir models. This indicates that fine-tuning enables the models to capture generalizable patterns in sequencing reads, rather than simply memorizing region-specific features. However, classification accuracy for HHV-8 drops sharply under this disjoint-read setup. In contrast, the performance on MCV, another low-abundance class, declines only slightly (less than 1%). Recall that while the MCV reads are even scarcer than HHV-8 reads, MCV genomes are over 20× shorter than HHV-8 genomes. These observations support the hypothesis that low sequencing coverage limits the learning ability of the NextVir framework, and hence its efficacy at classifying reads from classes sequenced at low coverage.

Among the models, NextVir-N performs particularly well in the context-supported setting, highlighting the robustness of the k-mer-based tokenization strategy introduced in [15].

## Robustness to mutations

Viruses are characterized by high mutation rates, leading to the continual emergence of novel strains and the formation of viral quasispecies [27–29]. The divergence from known viral sequences clearly presents a challenge to any viral identification or classification method. To evaluate the robustness of NextVir under these conditions, we construct a separate test set containing mutated viral reads, while keeping the training data unaltered. We simulate two forms of mutation in the viral sequencing reads—substitutions, where single nucleotide bases are changed, and indels, where bases are inserted or deleted. Computational approaches find indels particularly difficult to handle as a single base shift affects the reading frame of a given sequence. Substitutions, on the other hand, not only reflect biological variation but also serve as a proxy for sequencing errors. For example, the typical sequencing error rate for Illumina reads is approximately 0.1

**Table 3. Performance of context-supported classification with fine-tuned embeddings.**

|  | Accuracy (%) | | |
|---|---|---|---|
|  | NextVir-D | NextVir-N | NextVir-H |
| Overall top-1 | 86.85 | 89.38 | 79.89 |
| Human | 93.33 | 95.23 | 86.55 |
| HBV | 90.84 | 93.83 | 77.05 |
| HPV | 71.26 | 76.40 | 73.88 |
| HCV | 62.13 | 55.38 | 54.33 |
| EBV | 64.01 | 71.86 | 59.05 |
| HTLV | 79.43 | 77.55 | 75.09 |
| HHV-8 | 15.82 | 21.43 | 13.27 |
| MCV | 98.83 | 99.42 | 98.25 |

Table 4 summarizes the performance of the NextVir models under varying mutation rates. As expected, higher mutation rates consistently lead to greater degradation in classification accuracy across all classes. Nevertheless, all models are fairly robust to small levels of mutation; with a substitution rate of 5%, the overall accuracy decreases by less than 4% per model and the per-class accuracy for most viral classes decreases by less than 7%. Notably, the impact of mutations is disproportionately large for three classes: EBV, HHV-8, and MCV. These classes also happen to exhibit the lowest coverage and/or the lowest abundance, suggesting that improving coverage and representation for such classes may not only boost accuracy in standard settings (as we observed earlier) but also enhance robustness to mutations.

At high mutation rates, particularly if indels are present, all NextVir models exhibit a notable decline in classification performance. For NextVir-D, this vulnerability may stem from the Byte-Pair Encoding (BPE) tokenization scheme used by DNABERT-S. While BPE is generally resilient to substitutions, even a single base insertion or deletion can shift the sequence and drastically alter its tokenization. Although NextVir-N demonstrates strong robustness to substitutions, it too struggles with indels. In fact, the drop in classification accuracy for high-coverage HPV reads due to indels is nearly twice as severe for NextVir-N as it is for the other models. This supports the hypothesis that non-overlapping $k$-mer tokenization, as employed by Nucleotide Transformer, is rather sensitive to frame shifts introduced by indels. Surprisingly, the 1-mer tokenization used in HyenaDNA does not confer consistent robustness to indels in NextVir-H. Its performance deterioration typically falls between that of the other two models, despite the expectation that single-nucleotide tokenization would be more tolerant to small structural changes. Additionally, due to its lower baseline performance, NextVir-H is the weakest performer across all mutation levels.

## Robustness to contamination

Real-world DNA sequencing workflows may experience contamination from non-target DNA (e.g., bacterial or fungal sources). Moreover, growing evidence links certain bacterial species to cancer development across various tissues in the human body [30]. Although classifying

**Table 4. Robustness to mutations in viral sequencing reads.**

| Model | Top-1 | HBV | HPV | HCV | EBV | HTLV | HHV-8 | MCV |
|---|---|---|---|---|---|---|---|---|
| **Accuracy (Change in Accuracy)** | | | | | | | | |
| **Baseline - no mutation** | | | | | | | | |
| NextVir-D | 94.69 | 99.74 | 95.53 | 95.68 | 95.13 | 98.18 | 62.33 | 99.05 |
| NextVir-N | 93.90 | 99.78 | 96.75 | 95.67 | 89.03 | 98.48 | 60.47 | 100.00 |
| NextVir-H | 89.93 | 99.49 | 92.46 | 91.70 | 78.26 | 97.53 | 52.56 | 98.82 |
| **5% substitution** | | | | | | | | |
| NextVir-D | 91.72 (-3.0) | 93.44 (-6.3) | 91.20 (-4.3) | 91.05 (-4.6) | 78.44 (-16.7) | 89.94 (-8.2) | 52.09 (-10.2) | 84.02 (-15.0) |
| NextVir-N | 91.86 (-2.0) | 95.74 (-4.0) | 93.24 (-3.5) | 90.91 (-4.8) | 82.67 (-6.4) | 92.22 (-6.3) | 55.81 (-4.7) | 92.31 (-7.7) |
| NextVir-H | 86.77 (-3.2) | 92.11 (-7.4) | 88.12 (-4.3) | 86.16 (-5.6) | 70.21 (-8.1) | 87.10 (-10.4) | 46.98 (-5.6) | 79.88 (-18.9) |
| **10% substitution** | | | | | | | | |
| NextVir-D | 85.04 (-9.7) | 73.56 (-26.2) | 86.04 (-9.5) | 84.94 (-10.7) | 67.33 (-27.8) | 71.92 (-26.3) | 46.05 (-16.3) | 53.85 (-45.2) |
| NextVir-N | 86.88 (-7.0) | 82.78 (-17.0) | 86.82 (-9.9) | 83.24 (-12.4) | 75.00 (-14.0) | 83.30 (-15.2) | 56.74 (-3.7) | 72.19 (-27.8) |
| NextVir-H | 80.75 (-9.2) | 74.56 (-24.9) | 82.50 (-10.0) | 79.73 (-12.0) | 62.95 (-15.3) | 73.24 (-24.3) | 44.65 (-7.9) | 56.80 (-42.0) |
| **10% substitution + 5% indel** | | | | | | | | |
| NextVir-D | 76.16 (-18.5) | 48.04 (-51.7) | 76.72 (-18.8) | 78.26 (-17.4) | 54.64 (-40.5) | 51.99 (-46.2) | 49.30 (-13.0) | 23.67 (-75.4) |
| NextVir-N | 78.66 (-15.2) | 71.63 (-28.2) | 60.52 (-36.2) | 85.82 (-9.9) | 44.95 (-44.1) | 57.87 (-40.6) | 28.37 (-32.1) | 30.77 (-69.2) |
| NextVir-H | 74.96 (-15.0) | 59.52 (-40.0) | 74.31 (-18.2) | 78.14 (-13.6) | 50.26 (-28.0) | 55.22 (-42.3) | 43.72 (-8.8) | 29.59 (-69.2) |

bacterial sequences is beyond the current scope of NextVir, robustness of classifiers to such contaminant reads is essential for their real-world adoption. Evaluating this robustness also serves a dual purpose—it indicates whether NextVir can function effectively within a broader DNA mixture separation pipeline. In such a pipeline, the reads NextVir confidently predicts as non-oncoviral could be forwarded to downstream classifiers such as MetaTransformer or DeepMicrobes [31,32] to determine their bacterial origin.

To assess robustness to contamination, we generate reads from the human contaminome using the ART simulator [33]. New test sets are created by spiking the original test set with varying proportions of contaminant DNA—specifically, .5%, 1%, and 5%. These additional reads originate from bacterial and fungal species commonly encountered in human sequencing experiments and are labeled as part of the non-oncoviral or, for consistency, "Human" class. An ideal classification model should maintain high average accuracy in the presence of such contaminants, correctly identifying them as non-viral despite never encountering these sequences during training.

As illustrated in Fig 2, the overall accuracy of the NextVir models is only marginally affected by low levels of contamination and remains high even when 5% of the test reads originate from non-oncoviral, non-human sources. This robustness is particularly noteworthy given that, in real sequencing data, contaminant reads typically comprise just 1% of a sample [33]. The NextVir models prove to be robust across all contamination levels, with NextVir-D exhibiting the smallest drop in accuracy and NextVir-N the largest. These results indicate that, on average, NextVir is capable of handling realistic contamination scenarios, further supporting its utility in practical sequencing pipelines.

## Benchmarking against existing methods for binary classification tasks

To our knowledge, NextVir is the first deep learning scheme designed for multi-class oncoviral classification tasks; hence the results reported in the preceding sections show no competing benchmarks. In this section, we adapt NextVir to a binary classification setting (i.e., to the task of distinguishing viral from non-viral reads) in order to benchmark its performance against state-of-the-art approaches used in tumor virome analysis and metagenomics. To this end, the output layer of NextVir is modified to produce a single logit, with the label '1' indicating viral origin. Since the dataset is approximately balanced between viral and

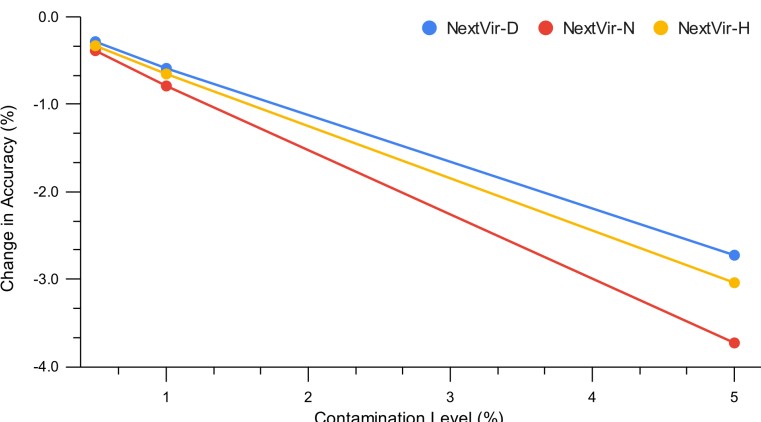

**Fig 2. Change in top-1 accuracy under various contamination levels.**

non-viral reads, binary cross-entropy loss is used for training. Competing methods include DeepVirFinder (DVF) [8], which employs a shallow convolutional neural network to detect bacteriophage DNA from metagenomic data; Virtifier [9], which combines an attention mechanism with an LSTM architecture; and XVir [34], a transformer-based method for viral read classification. For a fair comparison, all baseline models are retrained on the dataset used in this study, following the training protocols described in their respective papers. The primary benchmarking results are summarized in Table 5, and additional benchmarks are provided in S1 Appendix.

As shown in Table 5, the NextVir models perform competitively with the best existing binary classification method, DeepVirFinder (DVF), on the standard test set obtained via random splitting. Notably, they outperform all baseline methods in the more challenging context-supported setting, where training and test reads are drawn from disjoint genome segments. ROC curves for both test scenarios are presented in Fig 3. These results demonstrate the effectiveness of genomic foundation models in applications to viral read detection and classification tasks. In particular, the strong performance of NextVir under the context-supported split demonstrates its robustness to coverage gaps and sequence heterogeneity. This robustness suggests that the proposed approach may be well-suited for real-world sequencing applications where uniform genome coverage cannot be guaranteed.

## Discussion

We introduced NextVir, a novel framework for multi-class classification of oncoviral reads from next-generation sequencing data. Building on recent advances in genomic foundation models, including DNABERT-S, HyenaDNA, and NucleotideTransformer, NextVir leverages fine-tuning and adapter-based embedding modification to enable accurate classification of reads based on their origin. Our benchmarking on semi-experimental data demonstrates substantial gains from fine-tuning over frozen representations, highlighting the value of task-specific adaptation. NextVir proves to be effective and achieves high performance across a range of settings, including the challenging, practically-motivated scenario where the reads in the training and test sets originate from disjoint regions of the genome. Moreover, NextVir delivers state-of-the-art performance in the binary setting of viral versus non-viral detection. These results reinforce the promise of genomic foundation models as powerful starting points for deep learning solutions to not only the oncoviral read classification task, but a broader spectrum of problems in genomics as well.

The choice of foundation model in NextVir reflects inherent trade-offs between computational efficiency, sequence resolution, and robustness to genomic variation. In our evaluations, DNABERT-S achieves the second highest performance, with superior precision across most oncoviral species and high AUC-ROC scores in binary settings. We attribute this consistency to its species-aware embeddings learned through MI-Mix training. However, as

**Table 5. Performance of NextVir models, DeepVirFinder, XVir and Virtifier on the (binary) viral detection task.**

| Method | Random split | | Context-supported | |
| | Accuracy | AUCROC | Accuracy | AUCROC |
|---|---|---|---|---|
| NextVir-D | 98.22% | **0.999** | 91.55% | 0.985 |
| NextVir-N | 98.69% | **0.999** | **91.57%** | **0.989** |
| NextVir-H | 97.71% | 0.997 | 88.70% | 0.969 |
| DVF | **98.87%** | **0.999** | 89.90% | 0.968 |
| XVir | 98.00% | 0.998 | 88.50% | 0.968 |
| Virtifier | 93.71% | 0.984 | 71.70% | 0.922 |

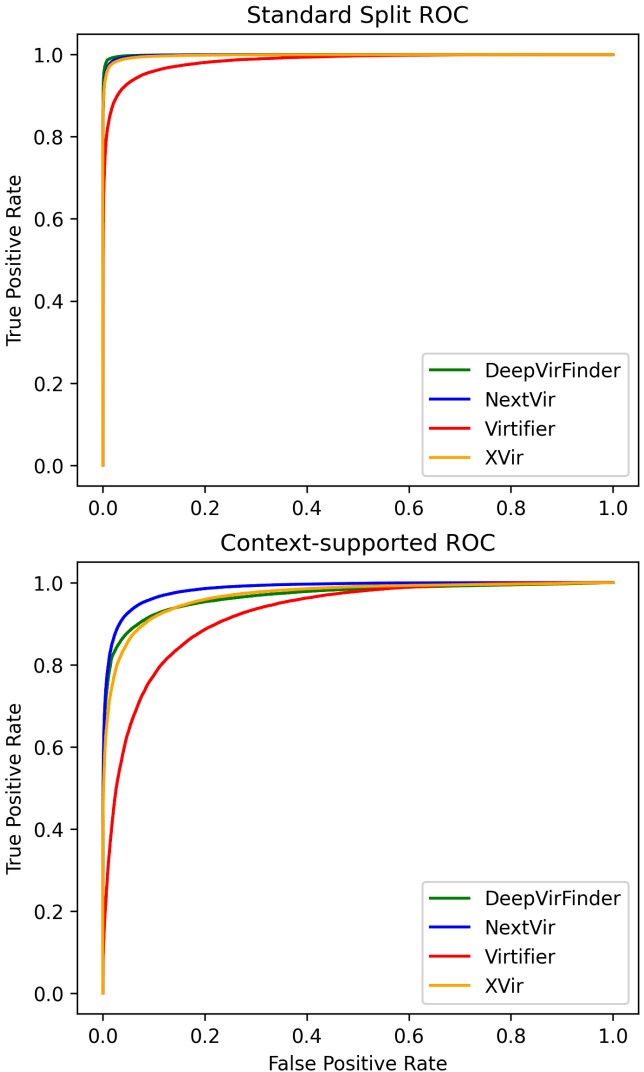

**Fig 3. Binary Receiver operating characteristic curves for NextVir-D, DVF, XVir and Virtifier on the test data in both randomly sampled and context-supported settings.**

discussed in section Robustness to mutations, DNABERT-S exhibits notable sensitivity to indels, which may affect its scalability in mutation-heavy or low-coverage settings. Nucleotide Transformer offers even stronger performance, particularly in substitution-robustness and out-of-distribution generalization, due to its multi-species pretraining and non-overlapping $k$-mer tokenization. Notably, NextVir-N yields the highest per-class accuracy among the pretrained (frozen) models, as reported in section Classification with pretrained embeddings. This contrasts with the findings in [17], likely reflecting our focus on oncoviral read-level classification. Still, its fixed $k$-mer encoding makes Nucleotide Transformer vulnerable to frame shifts introduced by indels (section Robustness to mutations), and it remains the most memory- and time-intensive model to train. HyenaDNA, with single-nucleotide resolution and a long-context processing architecture, offers the best computational efficiency and scalability, particularly for long-read inputs—results consistent with the conclusions

in [13]. However, this advantage comes at the cost of accuracy: NextVir-H consistently lags behind the other models in both multi-class precision and binary AUC-ROC scores, particularly in short-read classification tasks. Overall, our findings support the observation in [35] that the DNABERT-S architecture offers the most balanced trade-off among classification performance, robustness to mutations, and computational feasibility.

The clinical implications of NextVir extend beyond computational benchmarks, suggesting potential applications in precision oncology workflows. Accurate detection and classification of oncoviral content in tumor samples could enhance our understanding of viral carcinogenesis mechanisms and potentially inform therapeutic strategies. Existing clinical approaches often rely on targeted PCR-based detection methods that are limited to predefined viral types and may fail to identify co-infections or novel variants. In contrast, NextVir's ability to perform multi-class classification with high accuracy enables more comprehensive viral profiling, potentially revealing previously unrecognized viral associations across diverse cancer types. Moreover, the framework's robustness to contamination suggests it could be integrated seamlessly into existing clinical sequencing pipelines, without requiring specialized preprocessing or sample preparation protocols.

Despite the promising results, several challenges must be addressed before NextVir can be fully translated into clinical settings. Most notably, the observed performance degradation for viral classes with low sequencing coverage, such as HHV-8, underscores a fundamental limitation of sequence-based classification methods and highlights the need for further investigation. Additionally, while NextVir demonstrates reasonable robustness to substitution mutations, its performance deteriorates markedly in the presence of insertions and deletions, particularly for low coverage classes. Incorporating specialized alignment-based pre-processing or designing tokenization strategies that are inherently tolerant to indels could help mitigate this limitation in future iterations of GFMs. Lastly, expanding the viral reference database beyond the seven oncoviral families considered in this study to include newly identified or emerging oncogenic viruses would enhance the framework's applicability to viral discovery and broader clinical genomics.

## Supporting information

**S1 Appendix Contains additional sections detailing: Choice of LoRA rank; Single species binary detection; Context-supported classification; Optimization; Simulated viral discovery; Results on uniformly sampled data; Difficulty detecting HHV-8; Detecting HPV in experimental data; Additional benchmarks versus binary viral detection methods.** (PDF)

## Author contributions

**Conceptualization:** John Robertson, Shorya Consul, Haris Vikalo.

**Data curation:** John Robertson, Shorya Consul.

**Formal analysis:** John Robertson.

**Funding acquisition:** Haris Vikalo.

**Investigation:** John Robertson, Haris Vikalo.

**Methodology:** John Robertson, Shorya Consul, Haris Vikalo.

**Project administration:** Haris Vikalo.

 

**Resources:** John Robertson, Haris Vikalo.

**Software:** John Robertson.

**Supervision:** Haris Vikalo.

**Validation:** John Robertson.

**Visualization:** John Robertson.

**Writing – original draft:** John Robertson.

**Writing – review & editing:** John Robertson, Shorya Consul, Haris Vikalo.

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
