## [Decision Letter · Decision Letter 0]

22 Jan 2025

PCOMPBIOL-D-24-01878

NextVir: Enabling Classification of Tumor-Causing Viruses with Genomic Foundation Models

PLOS Computational Biology

Dear Dr. Robertson,

Thank you for submitting your manuscript to PLOS Computational Biology. After careful consideration, we feel that it has merit but does not fully meet PLOS Computational Biology's publication criteria as it currently stands. Therefore, we invite you to submit a revised version of the manuscript that addresses the points raised during the review process, in particular the problems of contamination that might occur in the dataset as pointed out by reviewer #1.

Please submit your revised manuscript within 60 days Mar 24 2025 11:59PM. If you will need more time than this to complete your revisions, please reply to this message or contact the journal office at ploscompbiol@plos.org. Please include the following items when submitting your revised manuscript:

We look forward to receiving your revised manuscript.

Kind regards,

Lin Hou

Academic Editor

PLOS Computational Biology

Arne Elofsson

Section Editor

PLOS Computational Biology

**Journal Requirements:**

At this stage, the following Authors/Authors require contributions: John Robertson. Please ensure that the full contributions of each author are acknowledged in the "Add/Edit/Remove Authors" section of our submission form.

4) We do not publish any copyright or trademark symbols that usually accompany proprietary names, eg ©,  ®, or TM  (e.g. next to drug or reagent names). Therefore please remove all instances of trademark/copyright symbols throughout the text, including:

- © on page: 1.

5) Your manuscript is missing the following section: Discussion.  Please ensure all required sections are present and in the correct order. Make sure section heading levels are clearly indicated in the manuscript text, and limit sub-sections to 3 heading levels. An outline of the required sections can be consulted in our submission guidelines here:

6) Please upload all main figures as separate Figure files in .tif or .eps format. For more information about how to convert and format your figure files please see our guidelines: 

7) We have noticed that you have uploaded Supporting Information files, but you have not included a list of legends. Please add a full list of legends for your Supporting Information files after the references list.

8) Please amend your detailed Financial Disclosure statement. This is published with the article. It must therefore be completed in full sentences and contain the exact wording you wish to be published.

2) State what role the funders took in the study. If the funders had no role in your study, please state: "The funders had no role in study design, data collection and analysis, decision to publish, or preparation of the manuscript.".

**Reviewers' comments:**

Reviewer's Responses to Questions

Reviewer #1: The manuscript by Robertson et al. presented a multi-class viral classification tool that adapts three genomic foundation models to detect oncoviruses from sequencing reads. The results showed that foundational models can be fine-tuned to perform viral classification at high accuracy, with performance comparable to well-established binary viral classifiers. However, several major issues should be carefully addressed before it can be published as a rigorous research paper, listed below.

1. NextVir is designed to identify oncoviruses from cancer cells. However, there are also diverse bacteria (Tekle et al., 2023) that are known or suspected to cause cancer. In addition, samples can be easily contaminated by environmental bacteria during sampling, processing, DNA extraction, library preparation, and sequencing. Thus, these scenarios should be considered during the design of NextVir or other similar tools.

2. The training dataset was semi-experimental, so it may not perfectly emulate actual sequencing data, particularly the diversity, highly variable coverage, and complexity of actual samples. Actually, the accuracy dropped significantly with the increase of mutations, and particularly indels, which are typical for real samples.

3. When benchmarking NextVir with other classifiers, the authors should A) use golden standard real-world metagenomes of human-associated, soil, marine, and other well-studied environments to capture the complexity of metagenomic samples. B) the authors should benchmark them at different read/contig lengths since the reads can be generated from different sequencing platforms, and it’s a common practice to detect viruses from assembled contigs, particularly those with long sequence lengths but low sequencing coverage. C) the authors should also compare the computational and memory costs when benchmarking these tools, which is vital to be adopted by the community.

Reviewer #2: The authors present a novel method for multi-class oncoviral read detection and classification utilizing next-generation sequencing data. The proposed method, NextVir, builds on several recent genomic foundation models, including DNABERT-S, NucleotideTransformer, and HyenaDNA, by fine-tuning and adapting their read embeddings to achieve accurate classification of reads based on their origin. Notably, NextVir also achieves state-of-the-art performance in binary viral detection tasks. Overall, the proposed framework has a superior performance and can solve not only the task of oncolytic virus read classification, but also various other problems in genomics.

1. In the Methods section, under Input Preprocessing, please provide a more detailed explanation of the padding method used and indicate whether it will have an effect on the final result.

2. The authors propose a NextVir approach based on three foundation models, DNABERT-S, NucleotideTransformer, and HyenaDNA. It is recommended to discuss the advantages and disadvantages of the three models in depth, describing the scenarios and potential limitations of each model.

3. It is recommended that the authors explore the possibility of extending the NextVir approach to other foundational models. This will help to assess the generalizability of NextVir and its potential for application in different genomics tasks.

4. It is recommended that the authors further discuss whether it is possible to further improve the performance of the model under low coverage conditions.

5. The authors mention that despite the scarcity of MCV reads, its genome length is much shorter than that of HHV-8, which may account for the more stable accuracy of MCV classification. It is recommended that the authors further explore the specific impact of genome length on model performance.

Reviewer #3: Reviewer’s comment:

Summary:

This paper presents NextVir, a novel framework for classifying oncogenic viruses using genomic foundation models. The study evaluates three advanced models—DNABERT-S, Nucleotide Transformer, and HyenaDNA—to distinguish between different viral genomes, including tumor-associated viruses such as HHV-8 and HPV-16. The models are tested on simulated and real genomic datasets, emphasizing their ability to handle indels, substitutions, and long-context dependencies.Key findings include DNABERT-S outperforming other models in accuracy and robustness due to its advanced fine-tuning with Low-Rank Adaptation (LoRA). Additionally, the paper highlights the computational trade-offs between accuracy and efficiency, with HyenaDNA showcasing speed advantages but struggling with certain viral classes. The robustness of NextVir to mutations and its potential application in metagenomic sequencing are discussed, positioning it as a promising tool for viral research and diagnostics. However, the study identifies limitations such as the reliance on simulated data and emphasizes the need for further validation on clinical samples.

Major comments:

1. While the use of DNABERT-S, Nucleotide Transformer, and HyenaDNA is explained, no comparison is provided against other potential foundational models. Discuss why alternatives like DeepVirFinder were excluded.

2. The reliance on simulated reads raises concerns about real-world generalizability. Clarify how well the simulated dataset mimics actual tumor samples and address potential biases.

3. Beyond accuracy, consider reporting F1 score, precision, and recall to provide a more comprehensive evaluation, especially for low-representation classes like HHV-8.

4. Discuss in more depth why some models are more robust to indels and substitutions. Relate this to tokenization schemes and architectural differences.

5. Include comparisons to simpler deep learning models (e.g., CNNs or LSTMs) trained on the same dataset to contextualize the performance gains of NextVir.

6. Clarify how the random and context-supported data splits were constructed and whether they accurately reflect sequencing scenarios in real-world studies.

Minor comments:

1. Tables: Include confidence intervals or standard deviations in accuracy results to highlight variability across runs.

2. Inconsistent verb tenses: In the "Results" section, the tense alternates between present and past. Ensure consistency by choosing either present (preferred for describing findings) or past tense.

**Have the authors made all data and (if applicable) computational code underlying the findings in their manuscript fully available?**

Reviewer #1: Yes

Reviewer #2: None

Reviewer #3: Yes

PLOS authors have the option to publish the peer review history of their article (what does this mean?). If published, this will include your full peer review and any attached files.

Reviewer #1: No

Reviewer #2: No

Reviewer #3: **Yes: **Jinhao Bi

**Figure resubmission:**
---

## [Decision Letter · Decision Letter 1]

19 Jun 2025

PCOMPBIOL-D-24-01878R1

NextVir: Enabling Classification of Tumor-Causing Viruses with Genomic Foundation Models

PLOS Computational Biology

Dear Dr. Robertson,

Thank you for submitting your manuscript to PLOS Computational Biology. After careful consideration, we feel that it has merit but does not fully meet PLOS Computational Biology's publication criteria as it currently stands. Therefore, we invite you to submit a revised version of the manuscript that addresses the points raised during the review process.

Please submit your revised manuscript within 30 days Aug 19 2025 11:59PM. If you will need more time than this to complete your revisions, please reply to this message or contact the journal office at ploscompbiol@plos.org. Please include the following items when submitting your revised manuscript:

We look forward to receiving your revised manuscript.

Kind regards,

Lin Hou

Academic Editor

PLOS Computational Biology

Arne Elofsson

Section Editor

PLOS Computational Biology

**Journal Requirements:**

1) Please upload the figures in the online submission form in a correct numerical order.

2) Please amend your detailed Financial Disclosure statement. This is published with the article. It must therefore be completed in full sentences and contain the exact wording you wish to be published.

1) State what role the funders took in the study. If the funders had no role in your study, please state: "The funders had no role in study design, data collection and analysis, decision to publish, or preparation of the manuscript.".

**Reviewers' comments:**

Reviewer's Responses to Questions

Reviewer #1: The manuscript has been comprehensively revised with additional supporting materials. I'm now satisfied with the current version and recommend acceptance as it is.

Reviewer #2: We appreciate the authors for revising the manuscript. Compared with the initial version, the manuscript has been significantly enhanced in multiple dimensions, particularly in the detailed elaboration of the NextVir model's performance, which makes the content presentation more coherent and clear. However, upon reviewing the revised manuscript, we identified that some issues remain inadequately addressed. To ensure the high quality of the article, we suggest the authors further refine it accordingly.

In the response letter, the authors stated that they had "dedicate the second paragraph of the new Discussion section to comparing the three foundation models used in our work". Nevertheless, the revised content remains at a general level of discussion without providing specific model evaluation metrics or citing authoritative scientific references. We recommend that the authors supplement concrete model evaluation indicators or incorporate relevant authoritative research when revising, thereby strengthening the objectivity and persuasiveness of the comparison.

Reviewer #3: The author answered my doubts.

**Have the authors made all data and (if applicable) computational code underlying the findings in their manuscript fully available?**

Reviewer #1: Yes

Reviewer #2: None

Reviewer #3: None

PLOS authors have the option to publish the peer review history of their article (what does this mean?). If published, this will include your full peer review and any attached files.

Reviewer #1: No

Reviewer #2: **Yes: **Peng Wang

Reviewer #3: No

**Figure resubmission:**
---

## [Editor Report · Decision Letter 2]

1 Jul 2025

PCOMPBIOL-D-24-01878R2

NextVir: Enabling Classification of Tumor-Causing Viruses with Genomic Foundation Models

PLOS Computational Biology

Dear Dr. Robertson,

Thank you for submitting your manuscript to PLOS Computational Biology. After careful consideration, we feel that it has merit but does not fully meet PLOS Computational Biology's publication criteria as it currently stands. Therefore, we invite you to submit a revised version of the manuscript that addresses the points raised during the review process.

Please submit your revised manuscript within 30 days Aug 31 2025 11:59PM. If you will need more time than this to complete your revisions, please reply to this message or contact the journal office at ploscompbiol@plos.org. Please include the following items when submitting your revised manuscript:

We look forward to receiving your revised manuscript.

Kind regards,

Lin Hou

Academic Editor

PLOS Computational Biology

Arne Elofsson

Section Editor

PLOS Computational Biology

**Journal Requirements:**

1) We have noticed that you have uploaded Supporting Information files, but you have not included a list of legends. Please add a full list of legends for your Supporting Information files after the references list.

**Reviewers' comments:**

**Figure resubmission:**
---

## [Editor Report · Decision Letter 3]

23 Jul 2025

Dear Mr. Robertson,

We are pleased to inform you that your manuscript 'NextVir: Enabling Classification of Tumor-Causing Viruses with Genomic Foundation Models' has been provisionally accepted for publication in PLOS Computational Biology.

Best regards,

Lin Hou

Academic Editor

PLOS Computational Biology

Arne Elofsson

Section Editor

PLOS Computational Biology

---

## [Editor Report · Acceptance letter]

PCOMPBIOL-D-24-01878R3

NextVir: Enabling Classification of Tumor-Causing Viruses with Genomic Foundation Models

Dear Dr Robertson,

I am pleased to inform you that your manuscript has been formally accepted for publication in PLOS Computational Biology. Your manuscript is now with our production department and you will be notified of the publication date in due course.

With kind regards,

Zsofia Freund
